

# Event-based stochastic point rainfall resampling for statistical replication and climate projection of historical rainfall series

Søren Thorndahl[1], Aske Korup Andersen[2], Anders Badsberg Larsen[2]

[1]Department of Civil Engineering, Aalborg University, Aalborg, DK9220, Denmark
[2]Niras A/S, Aalborg, DK-9000, Denmark

*Correspondence to*: Søren Thorndahl (st@civil.aau.dk)

**Abstract.** Continuous and long rainfall series are a necessity in rural and urban hydrology for analysis and design purposes. Local historical point rainfall series often cover several decades which makes it possible to estimate rainfall means at different time scales, and to assess return periods of extreme events. Due to climate change, however, these series are most
likely not representative for future rainfall. There is therefore a demand for climate projected long rainfall series, which can represent a specific region and rainfall pattern as well as fulfill requirements of long rainfall series which includes climate changes projected to a specific future period.

This paper presents a framework for resampling of historical point rainfall series in order to generate synthetic rainfall series, which has the same statistical properties as an original series. Using a number of key target predictions for the future climate,
such as winter and summer precipitation, representation of extreme events, the resampled historical series are projected to represent rainfall properties in a future climate. Climate projected rainfall series are simulated by brute force randomization of model parameters which leads to a large number of projected series. In order to evaluate and select the rainfall series with matching statistical properties as the key target projections, an extensive evaluation procedure is developed.

## 1 Introduction

In design of new and analysis of existing storm water drainage systems valid rainfall statistics are crucial. With climate changes anticipated to impact precipitation patterns, the historical rainfall statistics upon which the traditional design is based, is no longer valid for future design. There is therefore a need for climate projection of the rainfall statistics in order for these to represent the future loads on storm water drainage systems.

Traditionally many simple urban drainage systems are designed with Intensity-Duration-Frequency (*IDF*) –relationships, or
types of design storms (e.g. Unit Hydrograph: Sherman, 1932; Chicago Design Storm, CDS:  Keifer and Chu, 1957; SCS: NRCS, 1986) which represent statistics for rain with specific return periods. Climate projection of these types of design methods can be relatively simple e.g. by multiplying the design rain by a bias climate factor (e.g. Semadeni-Davies et al., 2008; Olsson et al., 2009; Willems et al., 2012a; Willems, 2013; Shahabul Alam and Elshorbagy, 2015) assuming that extreme rainfall events for a specific return period will be increased linearly with a given factor as a function of time. The
most recognized approaches for estimating climate factors is downscaling of Global Circulation Models (*GCMs*) or Regional Climate Models (*RCMs*), (e.g. Wilby and Wigley, 1997; Fowler et al., 2007).





In general, statistical downscaling determines a statistical relationship between a large- and a local-scale climate variable based on historical records. The relationship can be used on a *GCM/RCM* to obtain local variables for a specific domain in a given time frame of climate projection. (e.g. Wilby et al., 2002; Nguyen et al., 2007; Willems and Vrac, 2011; Willems et

al., 2012b; Arnbjerg-Nielsen, 2012; Sunyer et al., 2015). The statistical downscaling approach requires large historical records of observations in order to establish the necessary statistical relationships. Based on various types of statistical downscaling, climate factors for urban drainage design purposes (e.g for multiplication on *IDF*-relationships) can be derived by statistically comparing contemporary climate conditions with projected future rainfall with regards to specific return periods, and aggregation levels (durations) or rainfall (e.g. Mailhot et al., 2007; Larsen et al., 2009; Madsen et al., 2009;

Nguyen et al., 2009; Nguyen et al., 2010 ; Willems and Vrac, 2011; Olsson et al., 2012; Willems, 2013).

Whereas a large proportion of the recent research described above has been conducted on estimating climate factors for design purposes, there is also a significant need - not only to describe future extremes e.g. in the form of *IDF*-relationships, but also to be able to project climate changes to continuous rainfall time series. Basically, simple design methods assume agreement between the return period of the rain intensity (for a given duration), and on the other hand the return period of

the critical load in the drainage system (water level, flow, basin storage, etc.). Multiplying climate factors to simple design storms is sufficient for many applications of urban drainage design, however for more complex drainage systems with non-linear rainfall runoff response the simple design methods falls short. That is, for complex systems the return periods of the rainfall duration and intensity are not in agreement with the return periods of the corresponding drainage system state. Therefore, historical rainfall series (or climate projected rainfall series) are required for complex systems, in order to

estimate maximum water levels in manholes, flooding, to estimate the  return periods, and other loads on the drainage system such as outlet to recipient, inlet to wastewater treatment plants, combined sewer overflow, outlet flow, pollutants loads, etc. in the future climate (e.g. Schaarup-Jensen et al., 2009; Thorndahl, 2009; Thorndahl et al., 2015).

According to Willems et al., (2012a, 2012b) there are generally two methods that produce continuous climate projected time series either by *1) Stochastic rainfall generators* which generate locally representative synthetic rainfall conditioned on

climate variables in present and future climate; or *2) Resampling or weather typing methods,* in which future local rainfall is sought in historical rainfall records under equivalent historical climate conditions as projected in the future, or modified to represent future climate conditions.

In the literature, the most acknowledged methods for stochastically generating synthetic rainfall series are based on Poisson clusters processes and rectangular pulse models such as Bartlett-Lewis (Koutsoyiannis and Onof, 2001; Onof and Wheater,

1994, 1993; Segond et al., 2007; Onof and Arnbjerg-Nielsen, 2009; Paschalis et al. 2014; Kossieris et al. 2016) or Neyman-Scott (e.g. Entekhabi et al., 1989; Cowpertwait, 1991; Cowpertwait et al., 2002; Fowler et al., 2005; Burton et al., 2008; Cowpertwait, 2010; Paschalis et al., 2014; Sørup et al., 2016). Calibration of the generators is typically performed comparing generated series to observed series and adjusting relevant parameters prior to climate projection. Methods for estimating point rainfall (e.g. Cowpertwait et al., 1996; Marani and Zanetti, 2007; Onof and Arnbjerg-Nielsen, 2009) and spatially





distributed rainfall or multi-site generators with spatial dependency (e.g. Kilsby et al., 2007; Burton et al., 2008; Sørup et al., 2015) have been applied. These methods have been shown to provide likely results for hourly or daily time steps, but also have significant shortcomings in terms of modelling rainfall in a finer temporal resolution. For urban hydrological applications with fast rainfall-response, a temporal resolution of input data down to 1 to 10 minute is required (e.g. Schilling, 1991; Willems, 2000; Thorndahl et al., 2008; Thorndahl et al., 2016a, 2016b). Since, interested in maintaining the fine temporal resolution of observed rainfall series, generation of synthetic rainfall series using Poisson clusters is rejected as an applicable method.

*Resampling or weather typing methods* (Willems et al., 2012a, 2012b) of statistical downscaling outcomes of *RCMs/GCMs*, can provide data in the required temporal resolution, since directly based upon historical records. Arnbjerg-Nielsen (2012) applied historical rain series originating from another geographical region, which had a climate analogue to the projected climate in order to obtain continuous representative rainfall series for future climate conditions. Zorita and Von Storch (1999), Olsson et al. (2009), Willems and Vrac (2011), and Ntegeka et al. (2014) used historical records of rain and modified these records to represent climate-representative continuous climate projected rain series. Olsson et al. (2009) applied the *delta change method* to multiply historical records with bias climate factors depending on rainfall intensity levels in order to fit projections of extreme, seasonal and annual precipitation. This approach, however, was implemented without alternating the temporal variability and the seasonal distribution of events of the rain series and maintaining the chronology of the original series. This particular shortcoming might be problematic in order to project the frequency of extreme events sufficiently.

The approach presented in this paper is different from the methods presented above, although it can be considered as a variation of *resampling* combined with *stochastic generation*. Whereas other methods use other climate variables, e.g. pressure, temperature etc. as climate predictors, this approach aims at fitting statistical properties of climate projected precipitation directly. In this case, these properties are derived from other studies of RCM projection (see section 2 for details). The validity of the method therefore depends on whether the climate projected target variables are comprehensive and detailed enough to project the future rainfall upon. The aim is to develop a generally adaptive method which can be applied to an arbitrary rainfall series and with different climate scenarios and projection period. On the contrary to studies described above, climate projected time series are generated directly for urban drainage modelling purposes. The objective has been to develop a generally applicable method that can be used directly by practitioners and scientist within the field of urban drainage who do not necessarily have detailed knowledge on climate projection, RCM's, downscaling, etc.

The procedure is divided in two major parts: 1) *Resampling of a historical point rainfall time series* (*Method development*: section 3.1 and *Results and evaluation*: 4.1); *and 2) Climate projection of resampled time series* (*Method development*: section 3.2 and *Results and evaluation*: 4.2);

The essential concept of the method is, to stochastically generate a large number of either resampled historical series or climate projected series, and to evaluate the statistical properties of the generated series against a number of key target variables. Rather than optimizing for the best parameter fit, the basic concept is to sample parameters from broad uniform





distribution functions for each parameter and to either accept or reject each simulated series using a specified criteria. Repeating, this procedure for a large number of realizations of rainfall series, it is possible to select a number of rainfall series which has a satisfying statistical representativeness in comparison with historical series or climate projection targets. The evaluation procedure is inspired by the GLUE method (Beven and Binley 1992, Thorndahl et al. 2008) and is presented
in detail in section 3.3.

## 2 Data

The method assumptions and subjectivity is discussed in section 5 and in section 6 conclusions on this approach to climate projection of single point historical rainfall series are provided.

The development of the model is based on rain gauge data from Denmark and projection of Danish climate conditions, but
could easily be extended to other regions/countries of interest.

Specific statistical properties for the future precipitation in Denmark are necessary in order to climate project the resampled rainfall series. This is made possible through knowledge on future global climate changes, including publications compiled by IPCC (2013) in *Assessment Report 5* (*AR5*) based on recent scientific literature in the field of research and knowledge about climate change. The aforementioned report contains information regarding the updated scenarios: *Representative*
*Concentration Pathways (RCP)*. However, there are not currently a sufficient number of simulated regional models of the *RCP*-scenarios (ensembles) to stipulate climate change impacts of the future precipitation patterns in Denmark in the detail required in this paper (Christensen et al. 2015). The provisional solution is to apply a scenario from the *Special Report on Emissions Scenarios* (*SRES*) from *Assessment Report 4* (IPCC, 2007). In Olesen et al. (2014) the Danish Meteorological Institute has collected and processed data from the *ENSEMBLES* project (http://www.ensembles-eu.org/,
http://ensemblesrt3.dmi.dk/, Van der Linden and Mitchell, 2009; Boberg et al., 2010; Maule et al. 2013). The report includes projection of weather extremes (including precipitation) using the *SRES A1B*-scenario and is produced from an ensemble of 14 regional climate models in the *ENSEMBLES* project. The *RCM's* are simulated for 1990, 2050 and 2100, but in this case only the 2100 runs are applied. Table 1 presents annual and seasonal (summer and winter) precipitation increment in 2100 compared to the reference in 1990. Furthermore, the report specifies changes in other climate indices. In the context of
precipitation, the variables *events above 10 mm*, *events above 20 mm*, and *max. daily precipitation* are relevant (table 2). In this paper these three variables are used to climate-project the resampled rainfall series, as they are considered important with regards to urban drainage modelling. The linear increase from 1990 to 2100 of these three variables is also presented in table 2 and used later on as climate factors.

Besides the projections of Olesen et al. (2014) the Water Pollution Committee of The Society of Danish Engineers has
published a report (guideline no. 30), with recommendations for design of drainage systems considering climate change (WPC, 2014, background report: Gregersen et al., 2014). Based also on the climate simulations of the *ENSEMBLES* project,





the climate factors for drainage system design in Denmark are recommended (table 3). Design rainfall, e.g. *IDF-*relationships, with a specified return period should be multiplied by this factor.

It is well-known that the climate projections includes large uncertainties, it is however beyond the scope of this paper to quantify this uncertainty. Other scenarios than *A1B* could have been applied and when an ensemble of the *RCP* scenarios

will become available, the proposed method could easily be updated.

The rainfall series which are applied in this study has its origin in the network Water Pollution Committee of The Society of Danish Engineers. At present, the network consists of 145 tipping bucket rain gauges (DMI, 2014). The rain gauge no. 5047 located in Sulsted, North Jutland (lat: 57.17, long: 9.96) is applied since this is a station with a long recording time and few errors compared to other gauge records. The gauge has been in operation over a period of 34 years from 1979 to 2014, but

due to minor interruptions in the dataset, the effective length of the series is 32 full years. The interruptions do not affect the statistical calculations as these are excluded from the data before the calculations are performed. The time series of 1 min. values for the Sulsted rain gauge is shown in figure 1.

In the *WPC* rain gauge network the temporal resolution of data is 1 minute. The start time of an event is determined at the minute of the first tip of 0.2 mm. All events therefore have initial values equivalent a multiple of 0.2 mm/min (12 mm/h).

These initial values are easily identifiable in figure 1. The end of an event is specified when there is not registered a tip within an hour. Using this definition of events, the minimum inter-event time (time between events) will be 1 hour.

Using Danish rainfall data on a daily scale Gregersen et al. (2014) have been able to identify multidecadal climate oscillations (Ntegeka and Willems, 2008; Willems, 2013a) as well as climate related changes in precipitation patterns over the past 140 years. Nevertheless, since this paper is based on evidently shorter rainfall series, it is assumed that no

significant trends or climate changes in this period are present. The historical records from the Sulsted series are therefore assumed not to contain climate change.

## 3 Method development

The procedure of the method is divided in two sections the *resampling of historical rainfall series* (section 3.1) and the *stochastic climate projection of resampled historical rainfall series* (section 3.2). Since both methods involve random

selection of events and brute force randomization of parameters there is a need for a unique method to evaluate the generated series against target values. This evaluation method, which is also used to optimize and condition the climate projected rainfall series against target values, is inspired by the GLUE methodology (Beven and Binley, 1992). The basic concept is to generate a large number of rainfall series and evaluate whether each generated series should be accepted or rejected based on an empirical likelihood (performance) measure based on individual criteria for each target value. For the accepted generated

rainfall series a combined performance measure for each realization is calculated in order to find the rainfall series realization which in general fit the target values the best. This method is described in detail in section 3.3.





## 3.1 Historical rainfall series resampling

The objective is to create synthetic rainfall series resampled from a historical series such that the synthetic and the historical series have the same statistical properties. The first step is to divide the historical rainfall series into smaller parts in order to describe variability of intensities, event duration and time between events over the year. It is chosen to divide the series into

four seasons (winter: DJF; spring: MAM; summer: JJA; autumn: SON), although a finer division (e.g. monthly) could have been implemented. Since the target projections (table 1) are implemented in seasons, this is the one used. The summer precipitation in the synthetic rainfall series is thus generated based on statistics calculated for every summer period's precipitation in the historical rainfall series and correspondingly for the other seasons.

The generation (resampling) is based on:

1.  Statistics of the *inter-event* time (also referred to as *rainfall intermittency*, e.g. by Molini et al., 2001 and Schleiss et al., 2011) using the definition of events presented in section 2.

  2.  Sampling of rainfall events including original event durations and intensities randomly from the pool of historical rain events for each season.

The concept is outlined in Figure 2.

The *inter-event times* ($t_{ie}$) for each season are approximated by a *two-parameter Generalized Pareto distribution* (*GPD*):

$$f(t_{ie}) = \left(\frac{1}{\sigma}\right)\left(1 + k \cdot \frac{t_{ie}}{\sigma}\right)^{\left(-1-\frac{1}{k}\right)} \tag{1}$$

Where $\sigma$ is a scale parameter and $k$ is a shape parameter. The advantage of this function is its ability to describe the entire range of the data including the extremes (see example in figure 3).

Molini et al. (2001) applied a *Weibull distribution* to describe the *inter-event time* of rainfall events, however based on the

dataset, presented in this paper, the fit of the *Generalized Pareto distribution* outperformed, *Weibull* and other similar distributions.

As opposed to other rainfall generators which use a fixed time scale, (e.g. Furrer and Katz, 2008), the time is sampled discontinuously in this case.

The sampling of the events is an automated process with random selection of events from the pool of historical rainfall

events for each season. When sampling a specific event, the intensity sequence and consequently also the duration is maintained. Synthetic resampled time series are, therefore produced by random alternating sampling of the *inter-event times* and historical events from a specific season. It is possible to sample the same event more than once. The procedure is repeated until the length of the generated series corresponds to the length of the historical series or any other specified length. The number and the chronology of events are therefore different from season to season and from year to year.

A vital assumption here is that events from the historical series can be sampled independently. Depending on the meteorological conditions at the time of a specific event there might potentially, be some correlation to prior and posterior events due to short *inter-event times*. Extreme event statistics and development of *IDF*-relationships, in Denmark is also produced assuming independent events (Mikkelsen et al., 1998; Madsen et al., 2009) so in order to preserve this



methodology, no inter-correlation between events has been implemented in the presented approach. The potential problem could be overcome by accepting only inter-event time larger than a specified value and by that combining events.

## 3.2 Climate projection and stochastic resampling of rainfall series

The climate projected rainfall series is generated in three steps:

1.    The *inter-event time* for each season is sampled using the same procedure as described in the previous section, however, the parameters of the *GPD* for each season is implemented as stochastic variables and thus sampled randomly from a uniform distribution with fixed upper and lower boundaries. This allows for different distributions of *inter-event times* than the ones used in the resampling of historical series. In the climate projected series it is thus possible to accommodate for climate changes in seasonal precipitation and the distribution between small and large

events, by changing the number of events per season. As an example the method is able to accommodate a moderate increase of total summer precipitation, and at the same time a considerable increase in frequency and intensity of extreme events, with generally a lower number of total events in summer as a result.

2.    Rainfall events are sampled from the pool of historical events for each season in the same way as described in section 3.1. The duration of each event is not alternated under impact of climate change, since there is presently no

evidence that single events will become shorter or longer in the future. This is obviously a crucial assumption, but nonetheless the best current estimate, which also has been applied by e.g. (Olsson et al., 2009). The sampling of events is therefore done without alternating the events from the pool, other than multiplying by different climate factors as presented below.

3.    For each season climate factors are multiplied to different intervals of intensities on the minute scale. The climate

factors are sampled randomly from uniform distributions with fixed limits. For each projected rainfall series there is a different climate factor for each season and intensity level. The number of intensity intervals and corresponding climate factors are found as an empirical selection of intervals based on the error evaluation of the performance. During the development of the procedure different intensity intervals were tested. To certify a good representation of both small and large summer events a log-scale separation with the following intervals were chosen (in mm/h): 0-

6, 6-12, 12-36, 36-108, and 108-∞. Using the Sulsted rain series these intervals corresponds to percentiles of the 1-min. values corresponding to (in %) 91.83, 91.83, 6.51, 1.4, 0.28, and 0.02. The number of data points in each interval is thus very uneven. However, this subdivision has empirically proven to provide a good representation of both annual and seasonal precipitation as well as a good representation of extremes (see section 4.2). Unlike the delta change method, applied by Olsson et al. (2009), this method has very few intensity levels on which a climate

factor is multiplied. However, since this method is based on independent random sampling of every climate factor for each interval and season (4x5=20 in total), rather than a continuous function predicting the climate change factor, few intervals are required in order to minimize the total number of realizations.



The total number of random variables in the current setup is thus 20 combinations of climate factor and season plus 8 *GPD* parameters (2 for each season).

### 3.3 Evaluation and optimization procedure

The governing assumption behind the resampling procedure is that the resampled rainfall series should have the equivalent statistical characteristics as the historical series on a number of key target variables. The climate projected resampled series should therefore also have the equivalent statistical characteristics by means of a number of key target climate projections (as the ones presented in tables 1-3). It is not a necessity that the same target variables are used to evaluate resampled historical rainfall series and the climate projected series, but it is chosen to do so in this paper, in order to keep the evaluation procedures the same regardless of generating series which should statistically represent historical series or climate projected series. The key target variables are described in detail below:

1.  *Annual precipitation (ap).* This target variable is included as it is a measure of the total "mass"-balance. Since the individual years of the resampled and historical series are not directly comparable year by year, the mean of all years is applied as target variable.

2.  *Seasonal precipitation (sp).* The mean seasonal precipitation is applied as a target variable in order to ensure same distribution between seasons in the resampled series. The four target parameters are labeled *spwi*, *spsp*, *spsu*, *spau* corresponding to winter, spring, summer, and autumn precipitation respectively.

3.  *Number of events above 10 mm per day (n10mm).* This target variable provides a measure of the representation of extreme events.

4.  *Number of events above 20 mm per day (n20mm).* Same as no 3.

5.  *Maximum daily precipitation (mdp, as a mean of the maximum day for all years).* This target variable also certifies the representation of extreme events.

6.  *IDF-relationships.* The intensity-duration-frequency relationships are traditionally applied in design of urban drainage systems and are therefore relevant to include as a target variable. In accordance with table 3, it is chosen to use the mean rain intensity over a duration of 60 minutes for return periods of 2 and 10 years respectively as a target value. The two values are labeled *d60T2* and *d60T10* respectively.

The performance of each individual target variable is estimated using a simple ratio measure between the target value and the corresponding modeled value:

$$P_{i,j} = 1 - \frac{|T_i - M_{i,j}|}{T_i} \qquad (2)$$

$P_{i,j}$ is the individual performance parameter for target variable $i$ (as presented above) corresponding to realization $j$; $T_i$ is the target value; and $M_{i,j}$ is the modeled value of the target variable of the $j$'th realization. For the evaluation of the resampled series against the historical series, $T_i = H_i$, where $H_i$ is the value of the target variable of the historical series. With respect to





the evaluation of the climate projected rainfall series, where the target value is given by a climate factor (*cf*) multiplied by the target variable of the historical series:

$$T_i = cf_i \cdot H_i \tag{3}$$

thus the the performance measure is:

$$P_{i,j} = 1 - \frac{|cf_i \cdot H_i - M_{i,j}|}{cf_i \cdot H_i} \tag{4}$$

*P* can vary between *0* and 1, where *P=1* corresponds to a perfect fit.

In order for a simulated rainfall series to be accepted $P_{i,j}$ has to be larger than a specified threshold. For the resampled historical the acceptance criteria for the individual performance measures is fixed and have been chosen to $P_{crit,i}=0.85$, hence all ten individual performance measures should exceed this value in order for the realization to be accepted. This means, that if a target value of just one of the 10 target values deviates more than 15 % from the value of the historical series, the realization is rejected.

For the climate projected series, it is possible to estimate individual values of the performance using the standard deviations of the climate factors (*cf*) given in tables 1 and 2:

$$P_{crit,i} = 1 - \frac{2 \cdot \sigma_{cf,i}}{cf_i} \tag{5}$$

Assuming Gauss distributed target variables, we will thus accept values which are within the 95 % confidence intervals of the distribution of each target variable. The acceptance criteria of the performance measure will thus be different for each target variable depending on the uncertainty (standard deviation) related to that specific climate projection (cf. table 1 and 2). The acceptance criteria for the performance of each target variable are presented in table 7 along with climate factors and standard deviation for each variable.

The combined performance measure $P_j$ of each realization series (*j*) is estimated as:

$$P_j = w_{ap} \cdot P_{ap,j} + w_{n10mm} \cdot P_{n10mm,j} + w_{n20mm} \cdot P_{n20mm,j} + w_{mdp} \cdot P_{mdp,j} + w_{d60T2} \cdot P_{d60T2,j} +$$

$$w_{d60T10} \cdot P_{d60T10,j} \tag{6}$$

where $w_i$ is the weights of the individual performance measures and $\sum w_i = 1$ . The *annual precipitation* (*ap*) is estimated weighting the seasonal precipitation by

$$P_{ap,j} = w_{spwi} \cdot P_{spwi,j} + w_{spsp} \cdot P_{spsp,j} + w_{spsu} \cdot P_{spsu,j} + w_{spau} \cdot P_{spau,j} \tag{7}$$

The individual weights are presented in section 4.2 and table 7. One could argue that each season should be given the same weight, however since only projections of summer and winter presentation is provided in Olesen et al. (2014), it is chosen to be able to give more weights to specific seasons. The values are estimated empirically by result analysis and accentuation of the most important parameters.



## 4 Results and evaluation

### 4.1 Historical rainfall series resampler

The synthetic resampled series are generated with the same total length as the original historical series. In this case 32 years. The *inter-event times* for each season are sampled from the two-parameter Generalized Pareto distribution as detailed in

section 3.1. The estimated values for the *shape* and *scale* parameters are presented in table 4. By comparing the *scale* and *shape* parameter, it is evident that there is a significant difference for each season. Therefore, it is important that the *inter-event times* are sampled individually for each season to ensure a representative number of events in the resampled rainfall series compared to the historical rainfall series. Figure 3 exemplifies empirical cumulative distribution functions for summer *inter-events times* for the historical series and for the fitted *GP*-distribution of summer *inter-event times*. Furthermore, the

empirical distribution from the resampled series with the best combined performance measure is presented *(P_j=0.97)*. Even though there is a small divergence of the fitted *GPD* especially between inter-event times of 1 and 5 days, the resampled data represents an almost perfect fit to the historical data. The Generalized Pareto is therefore considered to be a reasonable choice of distribution function for random sampling of the *inter-event times*.

There is a stringent dependency between *inter-event times* and *number of events* in the rainfall series. In order to generate a

valid and representational resampled rain series, the *number of events* series should correspond somewhat to the *number of events* in the historical rainfall series. Table 5 therefore includes the mean and standard deviation of the *number of events* per year even though the *number of events* are not used a target variable for estimating the individual performances.

The resampling of the observed rainfall series is performed generating 10,000 different resampled rainfall series and assessing the performance of each generated series using the method described in section 3.3. Out of the 10,000 realizations

of simulated series, 113 (1.1 %) are accepted using the criteria of a minimum individual performance measure ($P_{crit,i}$) of 0.85. The fact that all 10 individual performance measures have to be larger than the acceptance criteria have shown to be a tough criterion to fulfill. Often one or two of the 10 has a slightly lower value and the realization is thus rejected. On average the accepted realizations has a combined performance measure ($P_j$) of 0.92 (ranging between 0.89 and 0.97). Figure 4 presents a bar plot (blue shades) of each of the target variables for the historical series, the one resampled series with the highest

combined performance measure, as well as the mean of the accepted resampled series (with uncertainty bounds indicating the minimum and maximum of the accepted series).

Generally there is a good agreement between the historical series and the accepted series on the target parameters with the highest weights, eq. 3: annual and seasonal precipitation. This is actually the case for the majority of the 10000 realizations, however the performance measures becomes rather low if the extreme events are not represented correctly in the resampled

series and is in that case rejected. The variability between the resampled series is only due to the randomness assembling events and *inter-event times* from the historical series since the *GPD*-parameters for each season are fixed corresponding to the fit (table 4). The rejection of resampled series is therefore often due to either sampling of too few or too many "extreme" events within a season.



In many situations, only the one resampled series with the highest performance measure is of interest. Table 5, therefore, lists target values of the historical series and the resampled series with the highest performance measure (best fit). Besides the best combined performance measure of *P=0.97*, the individual performance measures are given in the right column. In order not only to compare series on mean values, table 5 also presents standard deviations describing the year to year variability over the total length of the series. Generally there is a satisfactory agreement on both mean and standard deviations between the historical series and the "best" accepted resampled rainfall series.

To verify the representativeness of extreme rainfall, figure 5 presents *IDF*-relationships for the historical and "best" resampled series for return periods of 2 and 10 years respectively. Generally, there is a good agreement between the curves which verifies the resampling method. There is however a minor divergence for short durations of the 10 year return period. In general the longer the return period the larger divergence between the curves must be expected as a result of the random sampling of historical events in the generated series.

Figure 6 shows the time series of the "best fit" resampled time series.

The overall assessment of the previous evaluation indicates that the rainfall resampler can represent the historical rainfall series well. Due to the stochasticity of the sampling of *inter-event times* and rainfall events, there is obviously some variability from year to year and from series to series, but since none of the target variables are significantly biased, the overall performance of the resampler is accepted. Since it is possible to produce resampled rainfall series with the same statistics as the corresponding original historical series, the resampling algorithm will be applied to generate climate projected rainfall series in the following section.

## 4.2 Climate projected rainfall series

Figure 4 and table 7 provides results for the climate projected rainfall series. The target variables (*Climate projected historical*) are estimated using eq. 3 and is thus the mean values of the historical series of table 5 multiplied by the climate factors specified in table 7.

Since the climate projection of rainfall series involves randomization of not only the event assembling, but also randomization of *GPD*-parameters and climate factors for different intensity intervals for each season, the generation of rainfall series requires a larger quantity of realizations compared to the resampling of series described in the previous section. Therefore a total of 100,000 climate projected rainfall series are generated. The acceptance criteria implemented are however slightly different compared to the ones detailed in section 4.1. In the evaluation of climate projected rainfall series an individual acceptance criterion for each target variable is estimated using eq. 5. For the ten target variables the acceptance criteria range between 0.79 (*mdp*) and 0.90 (*spwi*) as presented in table 7. Due to one target variable namely the *Annual number of events above 20 mm per day (n20mm)* it has not been possible to produce any accepted realizations where all 10 target variables are accepted in one realization. Since the climate factor of this variable assume a value as high as 2.5 (table 2 and 6) it has not been possible to generate climate projected rainfall series with that large an increase in the number of



events with more than 20 mm of rain within a total of 100,000 realizations. The projection of 2 days per year with more than 20 mm rain in 1990 to 5 days in 2100 (table 2) seems somewhat unrealistic compared to projections of other variables included in this study. The target variable, *n20mm*, is thus omitted from the individual performance parameter evaluation in the selection of accepted and rejected realizations. It is possible that increasing the number of realizations might enhance the

probability of also accepting this variable. This is subject to further investigations.

With only 9 target variables left to fulfill the acceptance criteria, the total number of accepted realizations is 1301 (1.3 %). The reason that a larger percentage is accepted here than in the previous section is that the acceptance criteria is somewhat softer encountering the uncertainty of climate factors. On average the accepted realizations has a combined performance measure ($P_j$) of 0.91 (ranging between 0.85 and 0.96).

Table 6 presents the mean *GPD*-parameters for the accepted climate projected realizations for each season. Comparing to table 4 (in which the parameter assessment is based on fitting the historical data) it is clear that the parameter values obtained by random sampling and performance measure evaluation of climate projected series do not deviate significantly from each other. There are however some smaller differences worth mentioning. Parameters for winter are similar indicating that the distribution of *inter-event times* is similar in historical and climate projected series. The shape parameter for summer is lower

for the climate projected series indicating slightly larger inter-event times. This is consistent with the *A1B* scenario predicting fewer events in summer and a slight increase in the total summer precipitation (5 %); hence a larger quantity of large events is necessary to certify the total rain balance during summer.

As detailed in section 3.2, the climate projection of resampled series is based upon multiplication of random climate factors in fixed intensity intervals. The 12 mm/h has a special significance as it corresponds to one tip per minute, cf. the event

definition in section 2. There is therefore a relatively large quantity of the data assuming this value. This problem could be overcome by smoothing over a 5 - 10 minute period, however it is chosen to maintain the 1-minute resolution of the data in order to generate climate projected series with the same features as the original rain series.

The mean (±standard deviation) of summer climate factors for the accepted realizations are: 0-6 mm/h: 1.01 (±0.05), 6-12 mm/h: 1.00 (±0.06), 12-36 mm/h: 1.28 (±0.24), 36-108 mm/h: 1.18 (±0.21), and >108 mm/h: 1.27 (±0.26). This indicates

that the climate projected rainfall series will consist of summer precipitation in which it is mainly the intensities larger than 12 mm/h which are larger compared to the historical series. The results are similar for the other seasons (not shown).

Generally there is a good agreement of climate projected target variables (*Climate projected historical*) and corresponding values for the climate projected resampled series (red shades in figure 4). There is, however, slightly more deviation compared to the present-time simulations. This is as expected since the climate projection includes more random parameters

and complexity. For the accepted realization with the highest performance measure, *P=0.96* (figure 4 and table 7) there is a tendency for the target variables related to the extreme values to be marginally underestimated. This is inevitably a result of high weights given to the target values related to total annual and seasonal precipitation. By changing the weights it would be possible to obtain more equal extreme values, however potentially at the expense of a poorer fit of the accumulated precipitation values.





In figure 5 in which the *IDF*-curves for both historical and climate projected series are shown, the slight underestimation of extremes also shows for the 10 year return period, however an overestimation of the 2 year return periods on low durations. Since the total length of the series is 32 years, return periods larger than 10 years are not presented well, since they the associated with large uncertainties (see e.g. Thorndahl, 2009)

Figure 7 shows the time series of the "best fit" resampled time series.

The overall performance of the climate projection of resampled rainfall series is considered to be acceptable. The introduction of 28 random variables and the random assembling of rain events, obviously require many realizations in order to produce accepted rainfall series which has a satisfactory degree of agreement on all target parameters.

## 5 Discussion

The developed procedure obviously involves a large degree of subjectivity in the choice of processes and parameters to include. This section will discuss and argue for some of these choices.

The target variables have been chosen to represent both annual and seasonal precipitation as well as more extreme values. The choice of the ten specific target variables is closely connected to the fact that this is what is currently available for Danish future climate conditions. However, when other, and maybe more detailed target variables becomes available, it would be possible to redo the generation of climate projected rainfall series with other target variables - for example, when they become available for the RCP-scenarios. Another possibility could be to implement other durations and return periods than for 60 minute durations for 2 and 10 years respectively, in order to emphasize specific extremes further.

It is of utmost importance that the chosen target variables are representative for the future precipitation patterns and that they are comprehensive in that way that they cover both annual/seasonal variations as well as well as single events and the statistics related to these. In this paper, it is chosen only to include mainly yearly mean values of target parameters (except the target variables related to return periods), but it could also be relevant to apply the year to year variability as a target in itself, in order to certify the correct representativeness of the resampled series in comparison with the original historical series.

The weights applied in estimating the overall performance of resampled series are chosen in order to emphasize the accumulated precipitation values, but on the other hand not neglect the extremes. Other weights could have been applied. One could imagine that the weights were chosen according to the purpose of use of the resampled and climate projected series. If, for example, the series should be used as an input to an urban drainage model simulating overflow from combined sewer systems to a recipient, it would probably be most important to have a good representation of the precipitation (event) totals. On the other hand if the purpose was simulating surcharge or flooding of a drainage system, the representation of extremes would be more important.



The weights chosen are also closely related to the subdivision of intensity intervals on which climate factors are multiplied. In the present setup only five intervals for each season were applied, however, a subdivision with more intervals, might enable more precise prediction of extreme events and potentially better projection of the *Annual number of events above 20 mm per day (n20mm),* which was omitted from the performance measure due to poor representation in the climate projections. Finer intervals would however be expensive in terms of more needed realizations, since more random parameters will have to be added. An idea to overcome this could be to make a continuous function (which could be parameterized and randomized) describing the relationship between climate factor and intensity rather than the discrete subdivision suggested here.

The proposed method applies two major assumptions which are relevant to discuss here. The first assumption is that events are sampled independently for each season. With inter-event times down to 1 hour, this might constitute a problem in hydrological applications where the response time of the system in question is larger than 1 hour. Hence, coupled events might impact the hydrological system response. The second assumption is that the duration of events does not change under changed climate signals. It has presently not been possible to find evidence for this contention in the scientific literature on climate change. Both of the assumptions are subject to further investigations.

## 6 Conclusion

This paper presented a procedure to generate both statistically representative resampled rainfall series from original historical rainfall series as well as climate projected rainfall series, which includes the advantages in local historical rainfall series as well as projections on changes in rain patterns in the future climate.

The simulated rainfall series can represent the climate projected target variables and it is shown possible to produce rainfall series which does not only project accumulated seasonal precipitation, but also extremes in correspondence with the climate projection of the *A1B* scenario. The procedure is generic so when other climate scenarios and potentially other target variables for further precipitation patterns become available, the method will be able to adapt to these as well.

The procedure for generating resampled and climate projected rainfall series fulfills a need for having local representative rainfall series which are valid both for the present and future climate. The series can be applied directly as inputs to urban drainage models in order to analyze the loads on a drainage system, e.g. combined sewer overflow, surcharge, storage filling, flooding in the present and future climate.

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



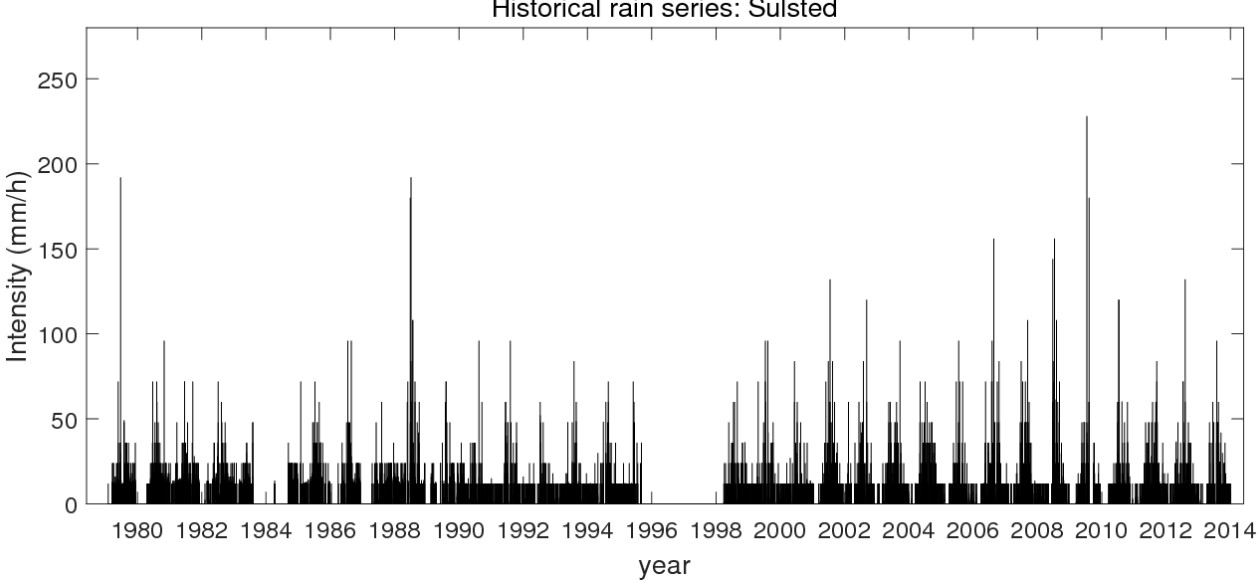

**Figure 1: Measured time series of the Sulsted rain gauge.**

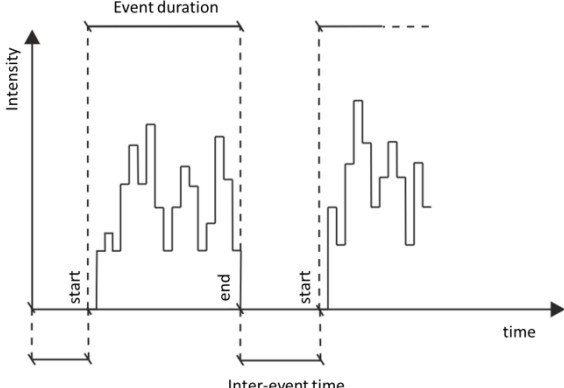

**Figure 2: Sketch of the construction of the synthetic (resampled) rainfall series.**




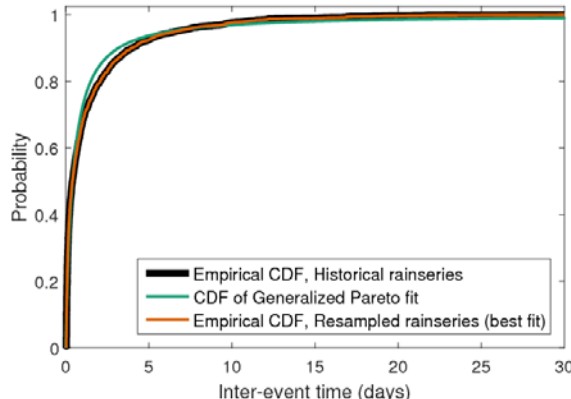

**Figure 3: Example of cumulative distribution functions for summer inter-event times**





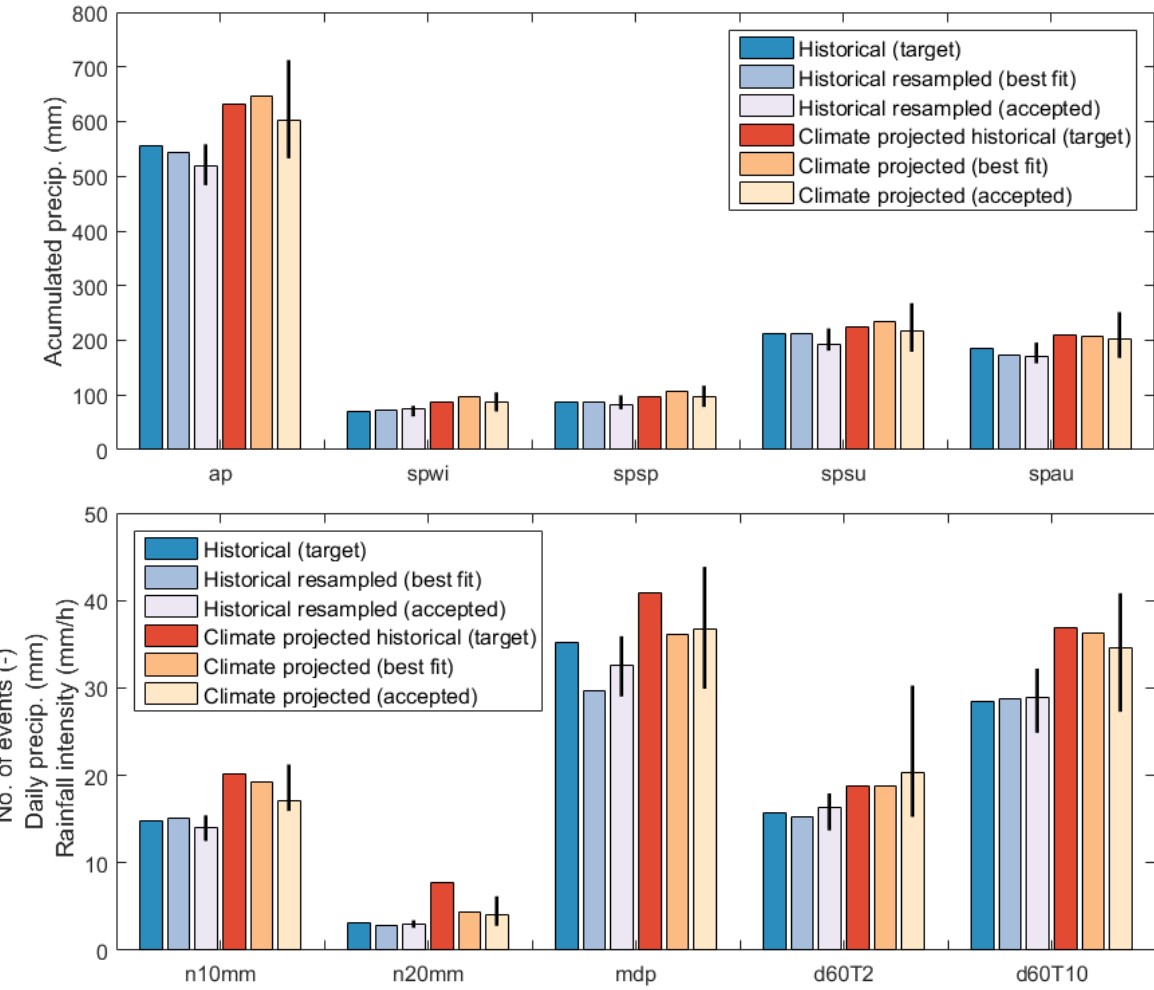

**Figure 4: Target variables and their values for comparing historical series and resampled series (in blue shades) and climate projected historical series and climate projected and resampled series (in red shades). The black lines indicate the range of the accepted realizations.**





**Figure 5:** *IDF*-curves for historical and resampled rainfall series respectively.

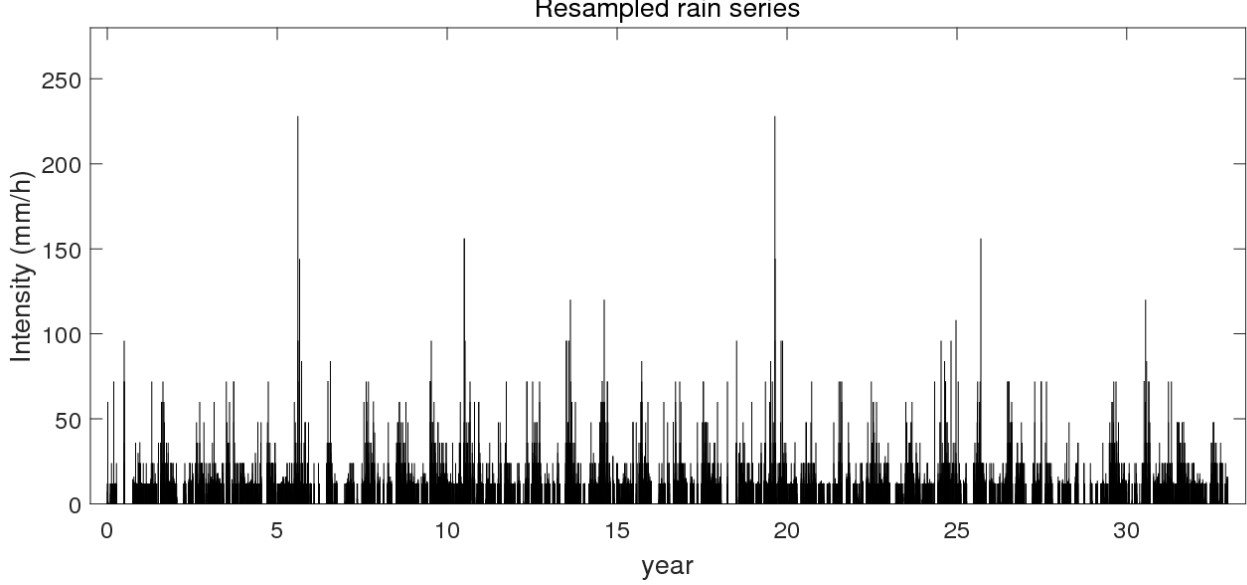

**Figure 6:** Time series example of resampled rainfall series.





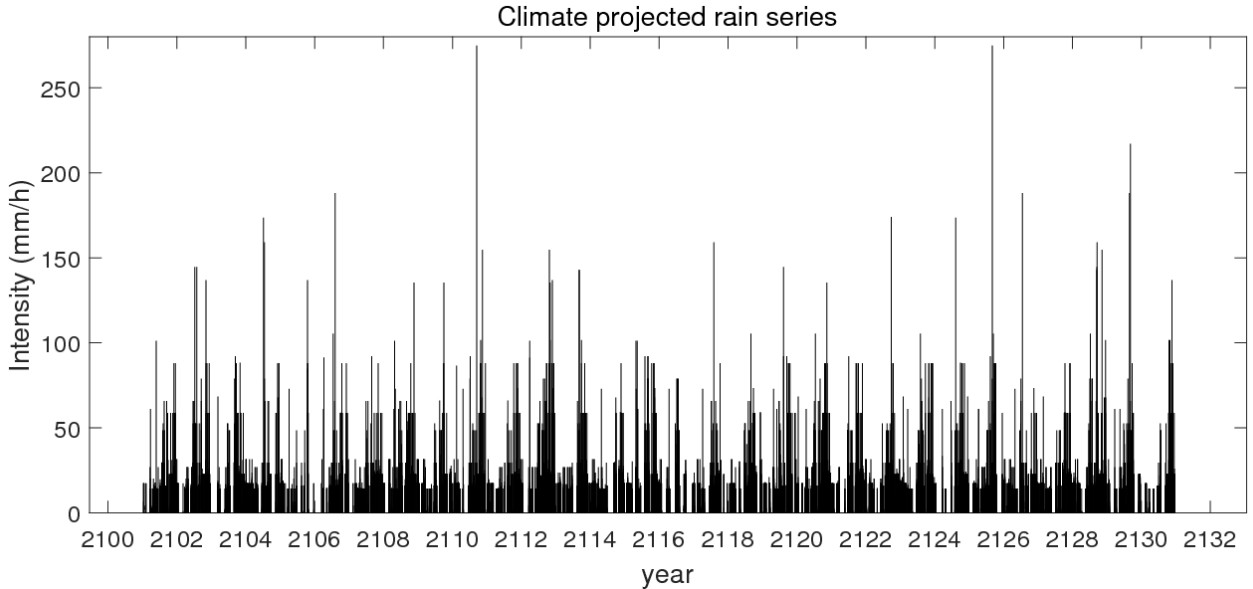

**Figure 7: Time series example of climate projected rainfall series.**

**Table 1: The calculated Danish climate change in annual, winter and summer precipitation expressed as percentage change in relations with the reference period 1961-90. The standard deviation is listed in parenthesis (Olesen et al., 2014).**

| Precipitation | Annual | Winter | Summer |
|---|---|---|---|
| Estimation (2100) | + 14% (± 6%) | + 25% (± 6%) | + 5% (± 8%) |

**Table 2 Climate parameters for future precipitation in Denmark using scenario A1B. The numbers represent the average of two periods; 1961-1990 and 2071-2100. The uncertainty indicates the standard deviation of the mean of 14 climate simulations (Olesen et al., 2014).**

| Indices | 1990 | 2100 | Precipitation change in DK |
|---|---|---|---|
| Events above 10 mm (days/year) | 19 (±2) | 26 (±3) | +37 % |
| Events above 20 mm (days/year) | 2 (±0.3) | 5 (±0.7) | +150 % |
| Max. daily precipitation (mm) | 70 (±8) | 81 (±10) | +16 % |




**Table 3: Recommended climate factors for design of drainage systems in Denmark according to WPC (2014) guideline no. 30, and Gregersen et al. (2014). The climate factors are valid for a duration of 1 hour, but also recommended for other durations. The indices marked with bold are the ones used in this paper. The standard deviations are not provided directly in the references, but estimated from tables and figures.**

| Return period (years) | Climate factor ($c_f$) |
|---|---|
| **2** | **1.2** (±0.1) |
| **10** | **1.3** (±0.2) |
| 100 | 1.4 (±0.3) |

**Table 4. The fitted scale and shape parameters for the Generalized Pareto distribution specified for each season.**

| Parameter | Winter | Spring | Summer | Autumn |
|---|---|---|---|---|
| Scale ($\sigma$) | 0.47 | 0.55 | 0.42 | 0.37 |
| Shape ($k$) | 0.97 | 1.01 | 0.89 | 0.82 |

**Table 5 Target variables (mean and standard deviation) and performance measures for the historical series and the one resampled series with the highest performance measure.**

| Target variable | | Acceptance criteria and weights | | Historical series (target) | "Best fit" resampled series | |
|---|---|---|---|---|---|---|
| | | $P_{crit,i}$ | $w_i$ | Mean (± Std. dev.) | Mean (± Std. dev.) | $P_i$ |
| Annual no. of events | | | | 200.1 (±39.4) | 197.4 (±55.5) | |
| Annual precipitation | ap (mm) | 0.85 | 0.5 | 555.2 (±122.3) | 543.1 (±170.2) | 0.98 |
| Seasonal precipitation, winter | spwi (mm) | 0.85 | 0.3 | 70.0 (±46.2) | 72.7 (±31.8) | 0.96 |
| Seasonal precipitation, spring | spsp (mm) | 0.85 | 0.2 | 86.5 (±43.1) | 86.5 (±45.8) | 1.00 |
| Seasonal precipitation, summer | spsu (mm) | 0.85 | 0.3 | 213.0 (±57.0) | 211.3 (±94.5) | 0.99 |
| Seasonal precipitation, autumn | spau (mm) | 0.85 | 0.2 | 185.7 (±53.6) | 172.6 (±80.9) | 0.93 |
| Annual number of events above 10 mm per day | n10mm (#) | 0.85 | 0.17 | 14.8 (±4.5) | 15.1 (±5.7) | 0.96 |
| Annual number of events above 20 mm per day | n20mm (#) | 0.85 | 0.08 | 3.1 (±2.1) | 2.9 (±1.7) | 0.96 |
| Annual Maximum daily precipitation | mdp (mm) | 0.85 | 0.08 | 35.2 (±12.7) | 29.6 (±9.5) | 0.93 |
| Rain intensity for 60 min, T=2 years | d60T2 (mm/h) | 0.85 | 0.08 | 15.7 | 15.3 | 0.98 |
| Rain intensity for 60 min, T=10 years | d60T10 (mm/h) | 0.85 | 0.08 | 28.4 | 28.8 | 0.99 |
| Combined performance measure | P | | | | | 0.97 |

**Table 6. Mean scale and shape parameters for the Generalized Pareto distribution of all accepted climate projected simulations specified for each season**

| Parameter | Winter | Spring | Summer | Autumn |
|---|---|---|---|---|
| Scale ($\sigma$) | 0.45 | 0.53 | 0.42 | 0.35 |
| Shape ($k$) | 0.97 | 0.98 | 0.80 | 0.78 |



**Table 7 Climate factors of target variables and minimum acceptance criteria of the individual performance parameters *Pi,j*.; empirical combined performance measure weights (w<sub>i</sub>); Climate projected target variables and the corresponding values (± standard deviation) of the best fit climate projected series.**

| Target variable | | Climate factors | Acceptance criteria and weights | | Climate proj. hist. series (target) | "Best fit" climate proj. series | |
|---|---|---|---|---|---|---|---|
| | | $c_f$ | $P_{crit,i}$ | $w_i$ | Mean | Mean (± Std. dev.) | $P_i$ |
| Annual no. Of events | | | | | | 216.9 (±58.6) | |
| Annual precipitation | *ap (mm)* | 1.14 (±0.06) | 0.89 | 0.5 | 632.9 | 618.5 (±180.3) | 0.98 |
| Seasonal precipitation, winter | *spwi (mm)* | 1.25 (±0.06) | 0.90 | 0.3 | 87.5 | 86.6 (±45.9) | 0.99 |
| Seasonal precipitation, spring | *spsp (mm)* | 1.13 (±0.10) | 0.82 | 0.2 | 97.7 | 100.51 (±48.7) | 0.97 |
| Seasonal precipitation, summer | *spsu (mm)* | 1.05 (±0.08) | 0.85 | 0.3 | 223.7 | 221.6 (±96.9) | 0.99 |
| Seasonal precipitation, autumn | *spau (mm)* | 1.13 (±0.10) | 0.82 | 0.2 | 209.8 | 209.8 (±85.3) | 1.00 |
| Annual number of events above 10 mm per day | *n10mm (#)* | 1.37 (±0.12) | 0.83 | 0.17 | 20.2 | 17.7 (±6.4) | 0.89 |
| Annual number of events above 20 mm per day | *n20mm (#)* | 2.50 (±0.14) | 0.89 | 0.08 | 7.7 | 3.9 (±2.04) | 0.52 |
| Annual Maximum daily precipitation | *mdp (mm)* | 1.16 (±0.12) | 0.79 | 0.08 | 40.7 | 37.2 (±13.1) | 0.99 |
| Rain intensity for 60 min, T=2 years | *d60T2 (mm/h)* | 1.20 (±0.10) | 0.83 | 0.08 | 18.8 | 21.3 | 0.87 |
| Rain intensity for 60 min, T=10 years | *d60T10 (mm/h)* | 1.30 (±0.20) | 0.85 | 0.08 | 36.9 | 34.6 | 0.98 |
| Combined performance measure | *P* | | | | | | 0.96 |

