# Peer review of "Event-based stochastic point rainfall resampling for statistical replication and climate projection of historical rainfall series"

_Hydrology and Earth System Sciences, 2017_

## Referee Comment (RC1) · P. Willems (Referee) · 26 May 2017

This paper describes a new approach for prolongation of rainfall series by resampling. The method can be applied, among other applications, on the basis of climate change impact analysis (statistical downscaling based on resampling). The approach is interesting but the paper needs strong revision. Some parts of the methodology are not fully clear, and the method has important limitations, which need further discussion.

Major comments:

Critical notes reg. the method:

Resampling from a (relatively short) historical time series has the disadvantage that

the same events may be taken several times. This may be problematic for the higher events; rainfall intensities for return periods longer than the length of the available historical times series will be underestimated by the proposed method. It will under-estimate the tail of the extreme value distribution. The same problem holds for any statistical downscaling method based on resampling. It therefore would be good to test the accuracy for the high return periods, which was not done.

Page 7 - lines 31-32: "rather than a continuous function predicting the climate change factor, few intervals are required ...": Again, this may lead to an approach that is "too deterministic" in view of the higher extremes (underestimation of the higher extremes). Why didn't the authors opt for a continuous function (could be a function with few parameters)? Is the monotonously change (growth) in change factor with increasing intensity interval guaranteed by the method proposed by the authors?

Page 7 - lines 14-15: "The duration of each event is not alternated under impact of climate change, since there is presently no evidence that single events will become shorter or longer in the future": There appears to be an inconsistency in the approach. By changing the parameters of the GPD for the inter-event time, the duration of the dry periods will change. At the same time the authors assume no change in the duration of the wet periods!?

On the clarity of the presentation:

Page 7 - lines 19-32: The method for applying the climatic change factors is unclear and confusing. As indicated in Table 3, the authors applied only change factors for events with return periods of 2, 10 and 100 years. Were only these factors used? Or, were also the seasonal change factors (Table 1) and the change factors in Table 10 used? At which time scale do the authors determine the return period of the event?

On the applicability of the method:

Page 10 - lines 19-20: "Out of the 10,000 realizations of simulated series, 113 (1.1 %)

[Figure]

are accepted": 10000 time series for 32 years and only 113 time series are accepted. Apparently, the stochastic process involves large uncertainties... How many time series would be needed for a 100-year historical time series?

On the accuracy of the method:

Figure 5: Given that the intensities are plotted on a log-scale, the differences between the historical and resampled values are very huge for some durations, esp. for the 10-year return period.

Figure 4: It is unclear what the vertical black lines for "Historical resampled (accepted)" and "Climate projected (accepted)" exactly represent. Does it represent (for the top figure) the +/- standard deviation intervals for the mean annual precipitation depth based on the 10000 series? Their length is in any case very large, esp. given that this is for the annual precipitation depth and for 30-year data. Same figure - vertical black lines for "Climate projected (accepted)": These lines are very large, esp. given that the climate model and RCP uncertainty are not explicitly considered and that the change factors are derived from a relatively small ensemble of 14 RCM runs only.

Figure 4: It is a big surprise that the uncertainty on the d60T10 (vertical black line shown for "Climate projected (accepted)" is relatively small, esp. in comparison with the uncertainty on the annual precipitation depth (top figure).

Table 5: It would be useful to evaluate as well the wet day frequency, and the duration of wet and dry spells. This would also provide more insight in the accuracy of the annual/seasonal precipitation depths. Table 5: Are the mean and standard deviation reported in the table based on all stochastic series (10 000)?

Other comments:

Introduction section - page 3: The authors make an overview of the different existing methods, but the overview is not fully correct. Lines 13-14 tend to indicate that the "delta change method" belongs to the class of "resampling and weather typing methods" which is not true. The method by Ntegeka et al. (2014) is the quantile perturbation methods which can be classified as an advanced "delta change method". Lines 15-16: "without alternating the temporal variability": this is not fully true. Willems and Vrac (2011) and Ntegeka et al. (2014) also change the number of dry and wet days in the time series.

Another drawback of the paper is that it makes use of the old generation RCMs and greenhouse gas scenarios (SRES; and only 1 SRES scenario was considered: A1B). Since several years, newer generation RCMs and greenhouse gas scenarios are available (EURO-CORDEX, CMIP5 based; RCP scenarios based). However, given that the paper focuses on presenting and testing a new methodology, this does not pose a real problem, but it is a pity that the older generation models and greenhouse gas scenarios were considered. Page 4 - lines 15-16: The authors claim that "there are not currently a sufficient number of simulated regional models of the RCP-scenarios ...": This is not true. The authors consider 14 RCM runs from the ENSEMBLES project, but 14 RCM runs (and even more) are also available in the EURO-CORDEX database (based on the RCP scenarios).

The inter-event times are described by the GPD distribution. It is a surprise that this distribution was selected. The GPD is typically valid and applied for extreme (POT/PDS) values (e.g. rainfall intensities), whereas inter-event times typically follow an exponential or Gamma type of distribution. Have these distributions (exponential, Gamma, two-component exponent) been tested? Page 6 - line 20: "outperformed": how was this evaluated? Page 6 - lines 20-21: "other similar distributions": which ones were tested? Figure 3: I suggest to change the plot where the vertical axis is log-transformed. If -ln(exceedance probability) of ln(return period) is plotted vertically, the deviation from the exponential distribution can be better evaluated (deviation from linearity). The current plot does not allow a proper evaluation of the goodness-of-fit.

Minor comments:

**HESSD**

Page 1 - Line 30: "... Global Circulation Models (GCMs) or Regional Climate Models (RCMs) ...": change "or" to "and/or"; approaches exist where climate factors are derived and applied from both GCMs and RCMs

Page 2 - line 5: change "large historical records" to "long historical records"

Page 2 - line 7: change "various types of statistical downscaling" to "various types of statistical downscaling assumptions and methods"

Page 3 - line 2: use a better word for "likely"

Page 3 - line 3: change "in a finer temporal resolution" to "at a finer temporal resolution"

Page 3 - line 4: change "1 to 10 minute" to "1 to 10 minutes"

Page 3 - line 5: change "Since, interested ..." to "Since we are interested ..."

Page 3 - line 6: change "is rejected as ..." to "is rejected here as ..."

Page 4 - lines 7-8: These two lines do not fit with this section about "Data".

Page 4 - line 25: It is unclear at this stage what exactly is meant by "events": number of events, or rainfall intensity of these events?

Page 5 - line 3: change "includes large uncertainties, it is ..." to "involve large uncertainties. It is ..."

Page 5 - line 6: change "the network Water Pollution Committee of ..." to "the monitoring network of the Water Pollution Committee (WPC) of ..."

Page 5 - line 21: change "assumed not to contain climate change" to "assumed to be stationary in terms of climate properties"

Page 5 - line 23: change "in two sections the resampling of ..." to "in two sections: the resampling of ..."

---

## Referee Comment (RC2) · L. Bengtsson (Referee) · 22 Jun 2017

The paper aims at determining rainfall for urban design. It is excellently written up to the result section, well describing previous research and the methodology used. However, the result section can be improved. When comparing observations and modeling it is not sufficient to conclude good or satisfactory agreemnet or letting the reader interpret figures by himself.

Since the main objective, besides deriving a new method for simulating rain series in a new climate, is to derive rain series that can be used for urban design, idf-curves should be shown in the conventional way as intensity vs duration for different return

periods. Although this is done in Fig. 5, the scale is not relevant. It would be better to use linear scale and not extent the duration further than 3 hours. I would like to see such curves directly after and based Fig. 1 and after Fig. 6. The two new figs should be compared and discussed more explicitly in the text.

Concerning climate change projections, I Think it should be told how large bias was used when improving the direct projections. As far as I know after simulating the present climate, correction factors are used to fit to observations and this bias is kept when modelling rain in future climate.

I agree with the authors that the expected increase of the number of 20 mm rain in a year seems unrealistic. I have done studies of several daily rains series extending more than 150 years and found significant increase of the number of 10 mm events, significant but minor increase of 20 mm daily events but none of 30 mm events. Perhaps the authors could look into long series of daily rainfall to investigate changes that have occurred.

A morec technical aspect is that the intensity in Fig 1 and 6 ought to be mm/min OR it should be clear from the legend that the graph shows rains over minutes. Also I Think the scale in Fig 3 should be changed. Usually log-scale is used or linear scale extending maybe up to 15 Days. On page 5 line 26 there is one value too much.

I am not sure whether my suggestions can be considered major revision or minor. I marked major.

---

## Author Comment (AC1) · 1 Jul 2017

The authors would like to express our gratitude to Patrick Willems for reviewing the paper. Willems is one of the most significant contributors to climate change projections in urban areas, and it is therefore an honor to have his view on our climate projection approach.

Below we will reply to his comments one by one. And argue how we will improve a revised version of the manuscript.

PW1: This paper describes a new approach for prolongation of rainfall series by resam-

pling. The method can be applied, among other applications, on the basis of climate change impact analysis (statistical downscaling based on resampling). The approach is interesting but the paper needs strong revision. Some parts of the methodology are not fully clear, and the method has important limitations, which need further discussion.

REPLY: We do actually not prolong the length of the series – but keep the length of the climate projected series the same length or shorter than the original series. This will be clarified in detail in lines 27-29 page 6.

PW2: Resampling from a (relatively short) historical time series has the disadvantage that the same events may be taken several times. This may be problematic for the higher events; rainfall intensities for return periods longer than the length of the available historical times series will be underestimated by the proposed method. It will underestimate the tail of the extreme value distribution. The same problem holds for any statistical downscaling method based on resampling. It therefore would be good to test the accuracy for the high return periods, which was not done.

Reply: We completely agree with the reviewers comment. It is indeed affecting the derived return periods if one extreme event is sampled more than once – and in our case this is probably also one of the reasons that we reject as many of the generated stochastic rainfall series as we do. The question of underestimating events with higher return periods than the total length of the series is indeed a problem, which is why we do not generate climate projected series longer than the total length of the original series. The problem on estimating return periods in the same range as the length of the series is also a problem for historical series, and this will not change by climate projecting series.

We will try to revise the text in order to explain clearer how we certify not to over- or underestimate the return periods. This will also be clearer by a revision of figure 5 as proposed in another comment and by another reviewer.

PW3: Page 7 - lines 31-32: "rather than a continuous function predicting the climate

change factor, few intervals are required ...": Again, this may lead to an approach that is "too deterministic" in view of the higher extremes (underestimation of the higher extremes). Why didn't the authors opt for a continuous function (could be a function with few parameters)? Is the monotonously change (growth) in change factor with increasing intensity interval guaranteed by the method proposed by the authors?

Reply: The separation in intervals was done in order to simplify the method, and keep the number of parameters at a minimum. But it is true that by parameterizing a continuous function of the change factor might actually entail fewer parameters. An approach to develop continuous functions will be investigated more in detail and if it shows plausible to change the method, we will do this in a revised version of the manuscript.

PW4: Page 7 - lines 14-15: "The duration of each event is not alternated under impact of climate change, since there is presently no evidence that single events will become shorter or longer in the future": There appears to be an inconsistency in the approach. By changing the parameters of the GPD for the inter-event time, the duration of the dry periods will change. At the same time the authors assume no change in the duration of the wet periods!?

Reply: This is indeed a crucial assumption of the method. The main idea is to maintain the chronology of each of the historical events also in the climate projected series. Since we apply relatively low thresholds for the minimum inter-event-time (down to 1 h), it is possible that two events in the climate projected series might be resampled with a short time in between and thus in practice be considered as one event.

This is indeed something that we will look into in the next generation of the proposed method. With more RCM ensembles in finer temporal resolution it might be possible to actually parameterize how duration of events will alternate in different climate scenarios, but this is at present not possible.

PW5: Page 7 - lines 19-32: The method for applying the climatic change factors is unclear and confusing. As indicated in Table 3, the authors applied only change factors

for events with return periods of 2, 10 and 100 years. Were only these factors used? Or, were also the seasonal change factors (Table 1) and the change factors in Table 10 used? At which time scale do the authors determine the return period of the event?

Reply: The change factors of table 3 was applied as target values for the 1h rainfall durations with 2 and 10 year return periods. We do actually not use the 100 yr return period. As it is the case with the other parameters, there are not related to any season, but to the high return periods. Since for durations of 1 hour – they are probably related to short summer events with high intensity. We will try to revise the text in order to make this clearer.

PW6: Page 10 - lines 19-20: "Out of the 10,000 realizations of simulated series, 113 (1.1 %) are accepted": 10000 time series for 32 years and only 113 time series are accepted. Apparently, the stochastic process involves large uncertainties... How many time series would be needed for a 100-year historical time series?

Reply: This is indeed a good question. In order to meet all our target variables, we reject a very large portion of the generated series. Since the resampling is performed completely random for each season it has some uncertainties in predicting all target variables. It is possible that the resampling could be optimized using other methods than brute force sampling, but since we are able to produce series which keep our defined criteria, the optimization of the method has not been a first priority. It is possible that this will be taken into consideration at a later point.

PW7: Figure 5: Given that the intensities are plotted on a log-scale, the differences between the historical and resampled values are very huge for some durations, esp. for the 10-year return period.

Reply: An effort will be made to quantify how large the differences are. Also as proposed by another reviewer figure 5 will be made clearer by dividing the figure in two. One for the historical series and one for the climate projected series and by limiting the plot to shorter durations.

**HESSD**

PW8: Figure 4: It is unclear what the vertical black lines for "Historical resampled (accepted)" and "Climate projected (accepted)" exactly represent. Does it represent (for the top figure) the +/- standard deviation intervals for the mean annual precipitation depth based on the 10000 series? Their length is in any case very large, esp. given that this is for the annual precipitation depth and for 30-year data. Same figure - vertical black lines for "Climate projected (accepted)": These lines are very large, esp. given that the climate model and RCP uncertainty are not explicitly considered and that the change factors are derived from a relatively small ensemble of 14 RCM runs only.

Reply: the vertical black lines are representing the total range of the accepted series (the ones which meet the target criteria). By accepting/rejecting more runs- the values would decrease/increase. Actually it is the subjective choice of target parameter values P, which determines the spread on the accepted realizations. We will clarify this in the text.

PW9: Figure 4: It is a big surprise that the uncertainty on the d60T10 (vertical black line shown for "Climate projected (accepted)" is relatively small, esp. in comparison with the uncertainty on the annual precipitation depth (top figure).

Reply: the uncertainty bounds are indicating the range of the d60T10 parameter in each of the accepted runs, and not the total uncertainty related to this parameter. It is thus the acceptance criteria of table 7 which controls the upper and lower limits.

PW10: Table 5: It would be useful to evaluate as well the wet day frequency, and the duration of wet and dry spells. This would also provide more insight in the accuracy of the annual/seasonal precipitation depths. Table 5: Are the mean and standard deviation reported in the table based on all stochastic series (10 000)?

Reply: Good idea. We will do an effort to include this in tables 5 and 7. The mean and standard deviation represents the mean and standard dev. from year to year in one series. The standard dev. between the stochastic series are only shown as the error bars in figure 4.

PW11: Introduction section - page 3: The authors make an overview of the different existing methods, but the overview is not fully correct. Lines 13-14 tend to indicate that the "delta change method" belongs to the class of "resampling and weather typing methods" which is not true. The method by Ntegeka et al. (2014) is the quantile perturbation methods which can be classified as an advanced "delta change method". Lines 15-16: "without alternating the temporal variability": this is not fully true. Willems and Vrac (2011) and Ntegeka et al. (2014) also change the number of dry and wet days in the time series.

Reply: We will clarify this in the manuscript

PW12: Another drawback of the paper is that it makes use of the old generation RCMs and greenhouse gas scenarios (SRES; and only 1 SRES scenario was considered: A1B). Since several years, newer generation RCMs and greenhouse gas scenarios are available (EURO-CORDEX, CMIP5 based; RCP scenarios based). However, given that the paper focuses on presenting and testing a new methodology, this does not pose a real problem, but it is a pity that the older generation models and greenhouse gas scenarios were considered. Page 4 - lines 15-16: The authors claim that "there are not currently a sufficient number of simulated regional models of the RCP-scenarios ...": This is not true. The authors consider 14 RCM runs from the ENSEMBLES project, but 14 RCM runs (and even more) are also available in the EURO-CORDEX database (based on the RCP scenarios).

Reply: After completing this manuscript we have received RPC-scenario ensamples (EURO-CORDEX) from DMI, which could be incorporated into this manuscript. We are planning to develop the method to be optimized by different RCP's, but in order to make this paper more clear on the development of the method, we will not include more than one RCP scenario.

PW13: The inter-event times are described by the GPD distribution. It is a surprise that this distribution was selected. The GPD is typically valid and applied for extreme

(POT/PDS) values (e.g. rainfall intensities), whereas inter-event times typically follow an exponential or Gamma type of distribution. Have these distributions (exponential, Gamma, two-component exponent) been tested? Page 6 - line 20: "outperformed": how was this evaluated? Page 6 - lines 20-21: "other similar distributions": which ones were tested? Figure 3: I suggest to change the plot where the vertical axis is log-transformed. If -ln(exceedance probability) of ln(return period) is plotted vertically, the deviation from the exponential distribution can be better evaluated (deviation from linearity). The current plot does not allow a proper evaluation of the goodness-of-fit.

Reply: We did test both exponential and gamma distributions, but found that the GPD represented both values with high and low probabilies better than the others. We might have a look at these distributions again in order to clarify this. We will change fig. 3 accordingly.

Closing remarks:

The minor comments will we taken into account revising the manuscript.

In conclusion. Other than minor revisions of figures, tables and text, we will improve the proposed method by considering and investigating the following three points:

1) Implementation of new RCP45 scenario rather than SRES A1B based on new EURO-CODEX ensamples. This might change some of the target variables slightly, but not change core of the method.

2) Implementation of the change factor as a continuous function of rain intensities rather than interval-based.

3) Have a closer look at probability functions for fitting the inter-event times. Might exponential or gamma functions provide better results than the GPD.
* * *
71, 2017.

---

## Author Comment (AC2) · 1 Jul 2017

The authors would like to pay our gratitude to reviewer L. Bengtsson for some useful comments and suggestions.

LB1: The paper aims at determining rainfall for urban design. It is excellently written up to the result section, well describing previous research and the methodology used. However, the result section can be improved. When comparing observations and modeling it is not sufficient to conclude good or satisfactory agreemnet or letting the reader interpret figures by himself.

[Figure]

Reply: We recognize that some figures needs to be clearer in order to support the conclusions and that the text should be elaborated in certain sections to underline the results more. We will revise the results section in order to make sure that readers of the paper will not have to interpret results themselves.

LB2: Since the main objective, besides deriving a new method for simulating rain series in a new climate, is to derive rain series that can be used for urban design, idf-curves should be shown in the conventional way as intensity vs duration for different return periods. Although this is done in Fig. 5, the scale is not relevant. It would be better to use linear scale and not extent the duration further than 3 hours. I would like to see such curves directly after and based Fig. 1 and after Fig. 6. The two new figs should be compared and discussed more explicitly in the text.

Reply: We agree that the IDF-curves are difficult to read. We will have a look at them again. Splitting them up in historical and climate projected as proposed might improve clarity, and we will have a look into if they can be well presented by a linear scale with max. 3h duration.

LB3: Concerning climate change projections, I Think it should be told how large bias was used when improving the direct projections. As far as I know after simulating the present climate, correction factors are used to fit to observations and this bias is kept when modelling rain in future climate.

Reply: This is s a good point. Currently this information is beyond our knowledge since climate model ensamples are executed and processed by DMI. But we will have a look at it.

LB4: I agree with the authors that the expected increase of the number of 20 mm rain in a year seems unrealistic. I have done studies of several daily rains series extending more than 150 years and found significant increase of the number of 10 mm events, significant but minor increase of 20 mm daily events but none of 30 mm events. Perhaps the authors could look into long series of daily rainfall to investigate

changes that have occurred.

Reply: We have not been able to investigate the orgin of this value from the data that was provided to us by DMI. Following reviewer Patrick Willems suggestion to apply new RCM ensamples based on RCP scenarios which has recently been available to us, we will have a closer look at the target parameter.

LB5: A morec technical aspect is that the intensity in Fig 1 and 6 ought to be mm/min OR it should be clear from the legend that the graph shows rains over minutes. Also I Think the scale in Fig 3 should be changed. Usually log-scale is used or linear scale extending maybe up to 15 Days. On page 5 line 26 there is one value too much.

Reply: All good points we will revise accordingly
* * *

---

## Author Response (AR1)

**31 July 2017**
**Søren Thorndahl**
**Aalborg University**

**Reply to reviewers and argumentation for changes in the original manuscript.**

Review by Patrick Willems

The authors would like to express our gratitude to Patrick Willems for reviewing the paper. Willems is one of the most significant contributors to climate change projections in urban areas, and it is therefore an honor to have his view on our climate projection approach. The comments have significantly improved the paper and the method especially the proposal for another distribution function for fitting inter-event times, a continuous function describing the change factor and the implementation of RCP scenarios rather than SRES.

Below we will reply to Willems comments one by one. And argue how we have improved the manuscript.

**PW1: This paper describes a new approach for prolongation of rainfall series by resampling. The method can be applied, among other applications, on the basis of climate change impact analysis (statistical downscaling based on resampling). The approach is interesting but the paper needs strong revision. Some parts of the methodology are not fully clear, and the method has important limitations, which need further discussion.**

REPLY: We do actually not prolong the length of the series – but keep the length of the climate projected series the same length or shorter than the original series. This will be clarified in the manuscript.

**PW2: Resampling from a (relatively short) historical time series has the disadvantage that the same events may be taken several times. This may be problematic for the higher events; rainfall intensities for return periods longer than the length of the available historical times series will be underestimated by the proposed method. It will underestimate the tail of the extreme value distribution. The same problem holds for any statistical downscaling method based on resampling. It therefore would be good to test the accuracy for the high return periods, which was not done.**

Reply: We completely agree with the reviewers comment. It is indeed affecting the derived return periods if one extreme event is sampled more than once – and in our case this is probably also one of the reasons that we reject as many of the generated stochastic rainfall series as we do. The question of underestimating events with higher return periods than the total length of the series is indeed a problem, which is why we do not generate climate projected series longer than the total length of the original series. The problem on estimating return periods in the same range as the length of the series is also a problem for historical series, and this will not change by climate projecting series. Since we are only considering ~30 years of rainfall we argue that we can only assess 10 year return periods.

We have revised the text in order to explain clearer that we do not over- or underestimate the return periods. This is also clearer by a revision of figure 5 as proposed in another comment and by another reviewer.

**PW3: Page 7 - lines 31-32: "rather than a continuous function predicting the climate change factor, few intervals are required ...": Again, this may lead to an approach that is "too deterministic" in view of the higher extremes (underestimation of the higher extremes). Why didn't the authors opt for a continuous function (could be a function with few parameters)? Is the monotonously change (growth) in change factor with increasing intensity interval guaranteed by the method proposed by the authors?**

Reply: The separation in intervals was done in order to simplify the method, and keep the number of parameters at a minimum. But it is true that by parameterizing a continuous function of the change factor might actually entail fewer parameters.

We have developed a new approach using a linear continuous function which shows significantly better results than in the original manuscript. It is evident by better performance measures as well as fewer realizations to obtain accepted runs.

**PW4: Page 7 - lines 14-15: "The duration of each event is not alternated under impact of climate change, since there is presently no evidence that single events will become shorter or longer in the future": There appears to be an inconsistency in the approach. By changing the parameters of the GPD for the inter-event time, the duration of the dry periods will change. At the same time the authors assume no change in the duration of the wet periods!?**

Reply: This is indeed a crucial assumption of the method. The main idea is to maintain the chronology of each of the historical events also in the climate projected series. Since we apply relatively low thresholds for the minimum inter-event-time (down to 1 h), it is possible that two events in the climate projected series might be resampled with a short time in between and thus in practice be considered as one event.

This is indeed something that we will look into in the next generation of the proposed method. With more RCM ensembles in finer temporal resolution it might be possible to actually parameterize how duration of events will alternate in different climate scenarios, but this is at present not possible.

**PW5: Page 7 - lines 19-32: The method for applying the climatic change factors is unclear and confusing. As indicated in Table 3, the authors applied only change factors for events with return periods of 2, 10 and 100 years. Were only these factors used? Or, were also the seasonal change factors (Table 1) and the change factors in Table 10 used? At which time scale do the authors determine the return period of the event?**

Reply: The change factors of table 3 was applied as target values for the 1h rainfall durations with 2 and 10 year return periods. We do actually not use the 100 yr return period. As it is the case with the other parameters, there are not related to any season, but to the high return periods. Since for durations of 1 hour – they are probably related to short summer events with high intensity.

We have changed tables 1-3 and the text accordingly in order to make this clearer. In this way there is more consistency between the tables. The RCP scenarios have also been introduced.

**PW6: Page 10 - lines 19-20: "Out of the 10,000 realizations of simulated series, 113 (1.1 %) are accepted": 10000 time series for 32 years and only 113 time series are accepted. Apparently, the stochastic process**

**involves large uncertainties... How many time series would be needed for a 100-year historical time series?**

Reply: This is indeed a good question. In order to meet all our target variables, we reject a very large portion of the generated series. Since the resampling is performed completely random for each season it has some uncertainties in predicting all target variables. It is possible that the resampling could be optimized using other methods than brute force sampling, but since we are able to produce series which keep our defined criteria, the optimization of the method has not been a first priority. It is possible that this will be taken into consideration at a later point.

However the replacement of the GPD to the mixed exponential distribution along with implementation of the continuous rather than the discrete change factors have significantly reduced the number of required runs to obtain satisfactory results.

**PW7: Figure 5: Given that the intensities are plotted on a log-scale, the differences between the historical and resampled values are very huge for some durations, esp. for the 10-year return period.**

Reply: We have changed figure 5 in order to make this more clear. We shorten the length of the x-axes and devide the figure in two. One for the present and on for the future climate. Furthermore the uncertainties have also been added to the plot.

**PW8: Figure 4: It is unclear what the vertical black lines for "Historical resampled (accepted)" and "Climate projected (accepted)" exactly represent. Does it represent (for the top figure) the +/- standard deviation intervals for the mean annual precipitation depth based on the 10000 series? Their length is in any case very large, esp. given that this is for the annual precipitation depth and for 30-year data. Same figure - vertical black lines for "Climate projected (accepted)": These lines are very large, esp. given that the climate model and RCP uncertainty are not explicitly considered and that the change factors are derived from a relatively small ensemble of 14 RCM runs only.**

Reply: the vertical black lines are representing the total range of the accepted series (the ones which meet the target criteria). By accepting/rejecting more runs- the values would decrease/increase. Actually it is the subjective choice of target parameter values P, which determines the spread on the accepted realizations.

This has been clarified in the text.

**PW9: Figure 4: It is a big surprise that the uncertainty on the d60T10 (vertical black line shown for "Climate projected (accepted)" is relatively small, esp. in comparison with the uncertainty on the annual precipitation depth (top figure).**

Reply: the uncertainty bounds are indicating the range of the d60T10 parameter in each of the accepted runs, and not the total uncertainty related to this parameter. It is thus the acceptance criteria of table 7 which controls the upper and lower limits.

Furthermore we have added also the uncertainty of the target implementing the standard deviations from tables 1 and 2.

**PW10: Table 5: It would be useful to evaluate as well the wet day frequency, and the duration of wet and dry spells. This would also provide more insight in the accuracy of the annual/seasonal precipitation depths. Table 5: Are the mean and standard deviation reported in the table based on all stochastic series (10 000)?**

Reply: Good idea. However we would like to limit the tables only to the target parameters only. Due to the change of the tables 1-3 in the original paper, as well as the changes in figure 4 it should be more clear what the standard deviations represent.

**PW11: Introduction section - page 3: The authors make an overview of the different existing methods, but the overview is not fully correct. Lines 13-14 tend to indicate that the "delta change method" belongs to the class of "resampling and weather typing methods" which is not true. The method by Ntegeka et al. (2014) is the quantile perturbation methods which can be classified as an advanced "delta change method". Lines 15-16: "without alternating the temporal variability": this is not fully true. Willems and Vrac (2011) and Ntegeka et al. (2014) also change the number of dry and wet days in the time series.**

Reply:  This has been clarified this in the manuscript.

**PW12: Another drawback of the paper is that it makes use of the old generation RCMs and greenhouse gas scenarios (SRES; and only 1 SRES scenario was considered: A1B). Since several years, newer generation RCMs and greenhouse gas scenarios are available (EURO-CORDEX, CMIP5 based; RCP scenarios based). However, given that the paper focuses on presenting and testing a new methodology, this does not pose a real problem, but it is a pity that the older generation models and greenhouse gas scenarios were considered. Page 4 - lines 15-16: The authors claim that "there are not currently a sufficient number of simulated regional models of the RCP-scenarios ...": This is not true. The authors consider 14 RCM runs from the ENSEMBLES project, but 14 RCM runs (and even more) are also available in the EURO-CORDEX database (based on the RCP scenarios).**

Reply:  After completing this manuscript we have received RPC-scenario ensamples (EURO-CORDEX) from DMI, which we have incorporated into this manuscript. We have done new simulations with the RCP4.5 instead of SRES A1B.

**PW13: The inter-event times are described by the GPD distribution. It is a surprise that this distribution was selected. The GPD is typically valid and applied for extreme (POT/PDS) values (e.g. rainfall intensities), whereas inter-event times typically follow an exponential or Gamma type of distribution. Have these distributions (exponential, Gamma, two-component exponent) been tested? Page 6 - line 20: "outperformed": how was this evaluated? Page 6 - lines 20-21: "other similar distributions": which ones were tested? Figure 3: I suggest to change the plot where the vertical axis is log-transformed. If -ln(exceedance probability) of ln(return period) is plotted vertically, the deviation from the exponential distribution can be better evaluated (deviation from linearity). The current plot does not allow a proper evaluation of the goodness-of-fit.**

Reply: We did test both exponential and gamma distributions, but found that the GPD represented both values with high and low probabilities better than the others. However revisiting this assumption we found

the the proposed two-component exponential (mixed exponential) provided much better results. We have thus implemented this throughout the paper.

Fig. has been changed accordingly.

The minor comments have been we taken into account revising the manuscript.

In conclusion: Other than minor revisions of figures, tables and text, we have improve the proposed method by implementation of following three points:

- Implementation of new RCP45 scenario rather than *SRES A1B based on new EURO-CODEX ensamples.* Implementation of the change factor as a continuous function of rain intensities rather than interval-based.
- Implementation of a mixed exponential distribution function instead of generalized pareto.

**Review by Lars Bengtsson**

The authors would like to pay our gratitude to reviewer L. Bengtsson for some useful comments and suggestions.

**LB1: The paper aims at determining rainfall for urban design. It is excellently written up to the result section, well describing previous research and the methodology used. However, the result section can be improved. When comparing observations and modeling it is not sufficient to conclude good or satisfactory agreemnet or letting the reader interprete figures by himself.**

Reply: We recognize that some figures needs to be clearer in order to support the conclusions and that the text should be elaborated in certain sections to underline the results more. We have revised the results section in order to make sure that readers of the paper will not have to interpret results themselves.

**LB2: Since the main objective, besides deriving a new method for simulating rain series in a new climate, is to derive rain series that can be used for urban design, idf-curves should be shown in the conventional way as intensity vs duration for different return periods. Although this is done in Fig. 5, the scale is not relevant. It would be better to use linear scale and not extent the duration further than 3 hours. I would like to see such curves directly after and based Fig. 1 and after Fig. 6. The two new figs should be compared and discussed more explicitly in the text.**

Reply: We agree that the IDF-curves are difficult to read. We have changed figure 5 significantly by splitting up in historical and climate projected and limiting to 3h durations. We did keep the log-scales however.

**LB3: Concerning climate change projections, I Think it should be told how large bias was used when improving the direct projections. As far as I know after simulating the present climate, correction factors are used to fit to observations and this bias is kept when modelling rain in future climate.**

Reply: This is s a good point. Currently this information is beyond our knowledge since climate model ensamples are executed and processed by DMI. We however think the changes in the original tables 1-3 will make the selected projections clearer to the reader.

**LB4: I agree with the authors that the expected increase of the number of 20 mm rain in a year seems unrealistic. I have done studies of several daily rains series extending more than 150 years and found significant increase of the number of 10 mm events, significant but minor increase of 20 mm daily events but none of 30 mm events. Perhaps the authors could look into long series of daily rainfall to investigate changes that have occurred.**

Reply: We have not been able to investigate the orgin of this value from the data that was provided to us by DMI. Following reviewer Patrick Willems suggestion to apply new RCM ensamples based on RCP scenarios which has recently been available to us, we don't experience the same problem. These issues have therefore been omitted from the paper.

**LB5: A morec technical aspect is that the intensity in Fig 1 and 6 ought to be mm/min OR it should be clear from the legend that the graph shows rains over minutes. Also I Think the scale in Fig 3 should be changed. Usually log-scale is used or linear scale extending maybe up to 15 Days. On page 5 line 26 there is one value too much.**

Reply: All good points we have revised accordingly.

[revised manuscript text omitted]
 t̶o̶ to WPC (2008), WPC (2014) g̶u̶i̶d̶e̶l̶i̶n̶e̶ ̶n̶o̶.̶ ̶3̶0̶, and Gregersen et al. (2014). The climate factors are valid for a duration of 1 hour, but also recommended for other durations up to 3 hours. The indices marked with bold are the ones used in this paper. The standard deviations are not provided directly in the references, but estimated from tables and figures.**

[revised manuscript text omitted]